# OpenReview forum: "Asynchronous Reasoning: Training-Free Interactive Thinking LLMs"
_ICML.cc/2026/Conference — Submitted to ICML 2026_

### Official Review · Reviewer_frKM · 2026-02-27

**Soundness:** 3
**Presentation:** 2
**Significance:** 3
**Originality:** 3
**Overall Recommendation:** 3
**Confidence:** 3

**Summary:**

This paper proposes an asynchronous reasoning method to make the LLM think and write concurrently.

**Compliance With Llm Reviewing Policy:**

Affirmed.

**Final Justification:**

This paper is generally interesting. My main concern is the completeness of the experimental evaluation. Overall, I believe it merits a score between 3 and 4.

**Key Questions For Authors:**

See weakness part.

**Limitations:**

yes

**Strengths And Weaknesses:**

Strength:
1. The focused problem is important.
2. The proposed method is interesting.
3. It is simple but useful.

Weakness:
1. The illustration of the proposed method is unclear. It is hard to understand how the writer and thinker collaborate.
2. The experiments have a "cherry-picking" problem. Different experiments use different base models. Not all experimental results from different models are reported. For example, Sec 4.4 only tests Qwen3-32B and Qwen3-235B-A22B-Thinking-2507.
3. As the paper claims that they are designed for real-time conversation, there is no experiment to test this scenario. I think a user study is needed to make this paper convincing.
4. It is still unclear how the method works. A case study is helpful to clarify this problem.

---

> ### Author Rebuttal · Authors · 2026-03-30
>
> Thank you for your review and suggestions for improving our work! We address your suggestions and conduct additional experiments below.
>
> > The illustration of the proposed method is unclear. It is hard to understand how the writer and thinker collaborate.
>
> Although we did our best with the intuition (Figure 1), thinker-writer view (Figure 2),  prompting scheme (Appendix A) and technical details (Figure 3), we are eager to further improve the explanations and add a case study.
> Here is an example of an enhanced Figure 1 (anonymized image: https://iili.io/BJBjqCP.jpg). We have shortened the example text and made some changes to color palette to improve understanding. The Figure 2 will be updated similarly in the final version of the paper. Additionally, we provide an animated visualization in our response to your suggestion #4 below.
> We will be glad to make further improvements if the reviewer will add any more comments about the figures.
>
> > The experiments have a "cherry-picking" problem. Different experiments use different base models. Not all experimental results from different models are reported. For example, Sec 4.4 only tests Qwen3-32B and Qwen3-235B-A22B-Thinking-2507.
>
> We respectfully insist that the presence of two models in Sec 4.4 is not cherry-picking: experiments for Figures 8 and 9 required running the same model on the entire dataset **7 times** for no delay and 6 different $k_{steps}$. Running additional models there took up a lot of GPU compute and complicated spacing constraints, so we only ran and reported two powerful models.
>
> That said, we agree that additional evaluations both in Sec 4.4 and other datasets would improve our paper. To that end, we report additional evaluations on MATH-500 Sharded and Lost in Conversation with sharded inputs for two more models. We use the same setup as in Section 4.4, the first column is k_steps, 0 means all shards are available right away.
>
> ### Table frKM.1: MATH-500 Sharded (accuracy)
>
> |k|Qwen3-8B|Qwen3-14B|
> |-|-|-|
> |0|89.0|92.8|
> |16|88.4|94.2|
> |32|88.6|94.0|
> |64|89.6|94.6|
> |128|89.2|92.8|
> |256|88.0|91.2|
> |512|79.6|82.0|
> |1024|48.6|62.6|
>
> ### Table frKM.2: Lost in Conversation math (accuracy)
>
> |k|Qwen3-8B|Qwen3-14B|
> |-|-|-|
> |0|73.8|80.6|
> |16|72.8|82.5|
> |32|75.7|69.9|
> |64|75.7|74.8|
> |128|70.9|68.9|
> |256|52.4|36.9|
> |512|20.4|8.7|
>
> Additionally, we report extra model-task pairs in our response to Reviewer 9yTv (Tables 9yTv.2, 9yTv.3) and additional comparisons in our response to Reviewer gTPM (Table gTPM.1). We appreciate the suggestion and will include these results beside figures 10-12 in the final version.
>
> > As the paper claims that they are designed for real-time conversation, there is no experiment to test this scenario. I think a user study is needed to make this paper convincing.
>
> A user study would indeed be interesting, even though it introduces complications. Since each user has a different speech cadence and habits on how soon they ‘interrupt’ the assistant, the results will vary significantly between populations, especially for dynamic conversations. For this reason, we chose to focus our experiments on automated latency tests. That said, it would indeed be interesting to measure real-world delays and cross-reference them with automated ones.
>
> To that end, we measure full end-to-end delay on a subset of MATH-500 samples on Qwen3-30B-A3B-Thinking-2507 in the same setup as in Section 4.2. This is the wall time from the moment the user stops talking to the moment the system begins speaking, and the frame-by-frame silence duration. Compared to our main evaluations, this includes both automated speech recognition and software / hardware delays.
>
> ### Table frKM.3: End-to-end latency evaluation for Qwen3-30B-A3B-Thinking-2507
>
> ||TTFT (s.)|Total Delay (s.)|
> |-|-|-|
> |AsyncReasoning (end-to-end)|4.643|9.887|
> |AsyncReasoning (Section 4.2)|3.901|9.145|
> |Baseline w/ think (end-to-end)|895.599|895.599|
> |Baseline w/ (Section 4.2)|894.884|894.884|
>
> The end-to-end delays are slightly (<1s) higher than the controlled evaluation from Section 4.2 due to speech recognition, input hardware and software delays. Note that the difference in total delays in our main results is much larger (tens to hundreds of seconds).
>
> >  It is still unclear how the method works. A case study is helpful to clarify this problem.
>
> Thank you for the suggestion! To address this, we created an animated view of AsyncReasoning in action, available at this anonymous link: https://freeimage.host/i/qpxKzJe . User voice inputs are omitted here for anonymity reasons. To better address your concern, we will include animated demonstrations of the full interaction loop in the final supplementary code.
>
> We hope we addressed the main concerns raised in your review with additional experiments, demonstrations and clarifications. If our response answers your questions, we kindly ask you to reevaluate your score. If you have further suggestions, we are happy to discuss them in the next phase.

---

> > ### Author Rebuttal · Reviewer_frKM · 2026-04-03
> >
> > Thank you for the rebuttal. I appreciate the added cases and figures, which make the proposed mechanism easier to understand. However, I do not think the concern about experimental completeness has been fully resolved. For example, Sec. 4.4 still does not include the gpt-oss family, and similar coverage issues remain in Sec. 4.2 and Sec. 4.3. I do not mean to suggest that all missing experiments should necessarily be completed within the limited rebuttal period. However, I believe that experimental completeness is important for adequately demonstrating the effectiveness of the proposed method, and this should have been addressed from the beginning. Overall, I find the paper interesting, but based on the current empirical evidence, I can only maintain my current score.

---

> > > ### Author Response · Authors · 2026-04-05
> > >
> > > Thank you for your response! Please find our answer to your concern below.
> > >
> > > > Sec. 4.4 still does not include gpt-oss
> > >
> > > We report these results below. Our previous response adds two more models to that setup, however, since our paper evaluates **10 models and 8 benchmarks**, we had to prioritize experiments. The additional results use the same shard insertion protocol as in Section 4.4, except that we use GPT-OSS reasoning format (Harmony). We report these new results in two tables below, for MATH-500 Sharded and Lost in Conversation math subset respectively. We keep the additional Qwen3 evaluations (Tables frKM.1, frKM.2) for your convenience and add two gpt-oss models.
> > >
> > > ##### Table frKM.4: MATH-500 Sharded (accuracy) **with gpt-oss**
> > >
> > > |k|Qwen3-8B|Qwen3-14B|gpt-oss-20b|gpt-oss-120b|
> > > |-|-|-|-|-|
> > > |0|89.0|92.8|87.8|93.6|
> > > |16|88.4|94.2|87.0|91.8|
> > > |32|88.6|94.0|88.4|92.2|
> > > |64|89.6|94.6|86.2|90.8|
> > > |128|89.2|92.8|82.0|88.2|
> > > |256|88.0|91.2|76.2|84.4|
> > > |512|79.6|82.0|55.8|77.6|
> > > |1024|48.6|62.6|39.6|48.2|
> > >
> > > ##### Table frKM.5: Lost in Conversation math (accuracy) **with gpt-oss**
> > >
> > > |k|Qwen3-8B|Qwen3-14B|gpt-oss-20b|gpt-oss-120b|
> > > |-|-|-|-|-|
> > > |0|73.8|80.6|72.8|82.5|
> > > |16|72.8|82.5|73.7|78.6|
> > > |32|75.7|69.9|70.9|79.6|
> > > |64|75.7|74.8|71.8|77.6|
> > > |128|70.9|68.9|63.1|70.9|
> > > |256|52.4|36.9|41.7|68.0|
> > > |512|20.4|8.7|14.5|13.6|
> > >
> > > Overall, the gpt-oss models behave similarly to our main results for Section 4.4, processing sharded inputs with minimal performance drops for smaller $k_{steps}$ and degrading for higher delays, which is also expected. The results for both gpt-oss models use **reasoning_effort = medium** (default, same as S4.2) for fair comparison, even though higher budget could further improve accuracy.
> > >
> > > > similar coverage issues remain in Sec. 4.2 and 4.3
> > >
> > > In our submission, we deliberately chose to evaluate in a wide range of setups to better understand and showcase asynchronous reasoning. Running all models in all setups is computationally expensive and, importantly, not all model / benchmark pairs are equally illustrative:
> > > 1. In 4.2, running larger models on MATH-500 and SpokenMQA does not fully utilize their reasoning, as our results demonstrate: see **Tables 9yTv.2** for 235B on MATH-500 and **9yTv.4** for SpokenMQA.
> > > 2. Likewise, we ran smaller models on hard benchmarks (e.g. AIME, see **Table 9yTv.3**), where AsyncReasoning still works about as well as the Thinking baseline, but all methods score poorly overall because these models are not capable enough for the challenging AIME-2025 benchmark.
> > >
> > > Still, we understand and agree with the desire to report the missing model pairs just to demonstrate that they are not anomalous. We already reported most of the requested models in **Tables 9yTv.2, 9yTv.3, 9yTv.4, gTPM.1 and TaQz.1** and will fill the missing results in the final version of the paper.
> > > If we understand correctly and this is indeed the only remaining concern, we gently request the reviewer to take these results into consideration.

---

### Official Review · Reviewer_TaQz · 2026-03-09

**Soundness:** 3
**Presentation:** 3
**Significance:** 3
**Originality:** 3
**Overall Recommendation:** 4
**Confidence:** 4

**Summary:**

The paper proposes AsyncReasoning, a training-free inference method that enables reasoning Large Language Models to think, process new inputs, and generate public responses simultaneously. It addresses the high latency of the traditional read-think-answer paradigm, which is a bottleneck for real-time applications like voice assistants and embodied agents. The method uses relative positional embeddings to create dual thinker and writer views within a shared key-value cache, avoiding redundant token computation. It employs a zero-shot mode-switching mechanism where the model self-determines when to pause output to wait for more reasoning. The authors evaluate the approach on math and commonsense benchmarks using models like Qwen3 and GPT-OSS, demonstrating significant reductions in time-to-first-token and overall delays while preserving reasoning accuracy. It also explores background safety checks and asynchronous handling of streaming inputs.

**Compliance With Llm Reviewing Policy:**

Affirmed.

**Final Justification:**

I retain my initial score.

**Key Questions For Authors:**

NA

**Limitations:**

yes

**Strengths And Weaknesses:**

### Strengths

1. The paper demonstrates that background reasoning can mitigate jailbreaks, dropping the Attack Success Rate (ASR) from 12.5% to 2.0% on HarmBench. This addresses critical safety-latency trade-offs in real-time systems.

2. The authors test diverse and challenging benchmarks including MATH-500, MMLU-Pro, GPQA-Diamond, AIME-2025, and ZebraLogic. This provides rigorous validation of general reasoning capabilities.

### Weaknesses

1. Smaller models, such as Qwen3-0.6B, lose significantly more accuracy with asynchronous reasoning compared to their synchronous thinking baselines. This limits the method's applicability on edge devices. The authors attribute this error to the writer giving answers prematurely, indicating that smaller models struggle with the zero-shot mode-switching prompt. This highlights a limitation in the training-free claim for lower-parameter models.

2. There is no detailed quantitative analysis of the failure rates of the mode-switching prompt across different model scales.

3. The analysis shows that varying T impacts both accuracy and TTFT/Delay, but a systematic way to determine the optimal T for a new task is not provided. This reduces the robustness of the method.

4. Typo: double "with" in line 432 of Section 5.

---

> ### Author Rebuttal · Authors · 2026-03-30
>
> Thank you for your feedback and suggestions. We are glad that your review highlights the diverse evaluations and the safety use case. We do our best to address your concerns below.
>
> > Smaller models, such as Qwen3-0.6B, lose significantly more accuracy with asynchronous reasoning compared to their synchronous thinking baselines.
>
> We agree with the raised concern and provide additional context below. In our training-free setting, smaller models do indeed lose more accuracy with AsyncReasoning, while relatively large models (>4B) work consistently. Upon closer inspection, this can be attributed to the fact that small models are worse at predicting when to pause and think and when to reply (see analysis below). This is a fundamental limitation of our training-free method, and we state that in L362-363.
>
> To adapt AsyncReasoning for local LLMs running on edge devices (e.g. smartphones or laptops), we see two potential strategies:
>
> 1. Compressing larger LLMs: practitioners from LocalLLaMA community previously ran quantized 8B models on mobile devices with quantization [1,2].
> 2. Training or augmenting smaller models for better routing, e.g. we could train an ad-hoc classifier to predict where Qwen-3-0.6B needs to pause and think more.
>
> To demonstrate the former, we evaluate on Qwen-3-8B-AWQ using the official 4-bit quantization [3]:
>
> ### Table TaQz.1 Quantized 4-bit Qwen3-8B-AWQ, MATH-500
> | |Accuracy|Avg. Delay (s.)|TTFT (s.)|
> |-|-|-|-|
> |8B-AWQ No Think|0.812|1.254|1.254|
> |8B-AWQ w/ Think|0.886|485.535|485.535|
> |8B-AWQ AsyncReasoning|0.88|6.351|1.809|
>
> > There is no detailed quantitative analysis of the failure rates of the mode-switching prompt across different model scales.
>
> We agree that a quantitative error analysis of mode switching would improve our work. To address this, we analyze the saved generations used in Figure 5. To separate mode-switching errors from other failure modes, we count specifically the cases where 1) the writer outputs a wrong answer prematurely before the thinker finishes (“Premature”), or 2) the thinker keeps generating and the writer never gets to the chance to write its answer because mode switching holds it back (“Stalling”). This covers both types of *critical* mode-switching errors that affect the final quality.
>
> ### Table TaQZ.2 mode-switching failure rates, Qwen3 models
> ||Premature|Stalling|
> |-|-|-|
> |0.6B|0.48|0.064|
> |1.7B|0.038|0.028|
> |4B|0.022|0.050|
> |8B|0|0.010|
> |14B|0|0.008|
> |32B|0|0.006|
>
> As we can see, smaller models do indeed make more frequent routing errors, which could explain why they often lose more accuracy in our evaluations. In the future work, it would be interesting to also quantify harder-to-find minor errors where the model could have begun answering slightly earlier. Thank you again for the suggestion, we will include this analysis in the final version of the paper.
>
> > The analysis shows that varying T impacts both accuracy and TTFT/Delay, but a systematic way to determine the optimal T for a new task is not provided. This reduces the robustness of the method.
>
> The analysis in Appendix E does indeed suggest that very large T (T>=50) result in unstable accuracy. In our submission, we use T=20 everywhere because we found that small deviations in T=5~25 range did not significantly affect accuracy, but we agree that Table 5 could have demonstrated that better. To address your concern, we extend Table 5 with additional Ts:
>
> ### Table TaQZ.3 Additional mode-switching rates (T), Qwen3-32B, MATH-500, A100
>
> | Frequency (T) | Accuracy | Delay(s) | TTFT(s) |
> |-|-|-|-|
> | 5 | 0.938 | 169.90 | 2.75 |
> | 10 | 0.936 | 117.82 | 3.34 |
> | 15 | 0.940 | 105.18 | 3.70 |
> | 20 | 0.938 | 102.10 | 4.32 |
> | 25 | 0.932 | 94.88 | 5.04 |
> | 30 | 0.922 | 83.09 | 5.12 |
> | 50 | 0.918 | 81.66 | 6.60 |
> | 100| 0.922 | 87.70 | 10.91 |
>
> To summarize, AsyncReasoning has consistent accuracy for T=5-25, but it becomes unstable as we increase T to larger values. On the other hand, using very small T (<=5) can make the model more responsive (TTFT), but it consumes extra compute to process the mode switching forward pass every T inference steps. As a result, we recommend using T in the 10-20 range as a sweet spot between consistency and compute overhead, but we agree that explaining this trade-off would be helpful to practitioners. We will include this in the final version of the paper.
>
> > Typo: double "with" in line 432 of Section 5.
>
> Thank you for pointing this out! We will correct this in the final revision.
>
> Overall, we hope that the new experiments and analysis alleviate your concerns. If you find these updates sufficient, please consider adjusting your score accordingly. If you have any follow-up questions or suggestions, we are glad to continue in the author-reviewer discussion phase.
>
> - [1] https://www.reddit.com/r/LocalLLaMA/comments/1clinlb/bringing_2bit_llms_to_production_new_aqlm_models/
> - [2] https://huggingface.co/ISTA-DASLab/AQLM-executorch
> - [3] https://huggingface.co/Qwen/Qwen3-8B-AWQ

---

> > ### Author Rebuttal · Reviewer_TaQz · 2026-04-01
> >
> > Thanks for the authors' reply. After reading the additional analysis, I think the score for this paper should be between 4 and 5, but I decide to keep a weak accept overall recommendation.

---

> > > ### Author Response · Authors · 2026-04-07
> > >
> > > Thank you for the update! If you have any further suggestions (please reply if so), we will be happy to include them in the final version of the paper.

---

### Official Review · Reviewer_gTPM · 2026-03-13

**Soundness:** 3
**Presentation:** 2
**Significance:** 3
**Originality:** 2
**Overall Recommendation:** 4
**Confidence:** 3

**Summary:**

This paper proposes AsyncReasoning, a training-free method that enables large language models to reason asynchronously rather than following the standard rigid read-think-answer cycle. Instead of forcing the model to finish all reasoning before responding, the method runs three concurrent token streams—user input, private thoughts, and public response—so the model can think while speaking and can also incorporate new user inputs during ongoing reasoning. The approach is implemented by modifying inference-time attention/cache behavior, using properties of rotary positional embeddings so that existing reasoning LLMs can support this behavior without retraining. Experiments suggest that AsyncReasoning reduces time to first token and user-perceived latency substantially while preserving the benefits of reasoning, and the paper also argues that it can help with interactive safety by allowing background deliberation to pause potentially harmful outputs.

**Compliance With Llm Reviewing Policy:**

Affirmed.

**Final Justification:**

Rebuttal solved most of my concerns

**Key Questions For Authors:**

Please refer to weaknesses

**Limitations:**

yes

**Strengths And Weaknesses:**

Strengths:

1.The paper identifies a real limitation of current reasoning LLMs: long internal reasoning improves quality but hurts interactivity. This is an important problem for voice assistants, interactive agents, and real-time systems.

2.A major strength is that the method is training-free and designed to work with existing reasoning models. That makes the idea practically attractive, since it avoids costly retraining and could be easier to integrate into existing inference stacks.

3.The paper goes beyond static reasoning benchmarks and connects the method to several realistic use cases, including streaming interaction, user interruptions/clarifications, and safety monitoring. This makes the contribution feel more impactful than a purely theoretical decoding change.

Weaknesses:

1.the contribution appears closer to an inference-time parallelization or scheduling strategy than to a new method for improving reasoning itself. The paper would be stronger with more direct comparisons to other approaches that switch between thinking and responding, such as adaptive deliberation, early-exit/streaming reasoning, or methods that explicitly control when the model should pause, think, or answer.

2.the effectiveness of AsyncReasoning may depend heavily on the post-training data format and the extent to which the base model has already been exposed to reasoning-style or multi-mode interaction patterns. If so, the method may not be as model-agnostic as claimed. More evidence on a broader range of models with different post-training procedures would help clarify whether the gains are truly general or mainly specific to certain reasoning-tuned models.

3.there is still a nontrivial drop in task performance in some settings. This raises an important practical question: how should users trade off reasoning quality against responsiveness? The paper would benefit from a more systematic study of this trade-off, along with possible mitigation strategies, such as adaptive pausing policies, confidence-based switching, or selective asynchronous reasoning only on easier segments of the interaction.

---

> ### Author Rebuttal · Authors · 2026-03-30
>
> Thank you for your feedback! We are glad you appreciated the simplicity and practicality of our approach and the real-world impact of interactive LLM reasoning. We do our best to address your concerns below.
>
> > The paper would be stronger with more direct comparisons to other approaches that switch between thinking and responding, such as adaptive deliberation, early-exit/streaming reasoning, or methods that explicitly control when the model should pause, think, or answer.
>
> We agree that additional comparisons with early-exit [1,2], adaptive deliberation [3,4,5] would strengthen our claims. To that end, we evaluate DEER [1] early exit and ThinkSwitcher [3] for adaptive thinking in the setup from Section 4.1 and official parameters [1,3]. Note that while DEER is training-free, ThinkSwitcher requires training a router for adaptation. We adapt the original Qwen3-8B setup and train MLP to choose between thinking and no thinking using MATH training splits (not MATH-500) and GSM8K, same as in the original paper [3]. We also evaluate ThinkSwitcher on top of our method, routing to AsyncReasoning or no reasoning.
>
> ### Table gTPM.1: MATH-500, Qwen3-32B, A100
> ||Accuracy|Avg.Delay(s)|TTFT(s)|
> |-|-|-|-|
> |DEER (early exit)|0.894|294.5996|294.5996|
> |ThinkSwitcher|0.936|451.7048|451.7048|
> |Baseline (Non-Thinking)|0.842|1.6326|1.6326|
> |Baseline (Thinking)|0.944|649.935|649.935|
> |Interleaved Thinking|0.874|16.244|4.011|
> |Interleaved Thinking (-Δ=0.5) |0.924|221.61|4.8832|
> |AsyncReasoning (main)|0.938|102.1|4.3213|
> |AsyncReasoning (speed prompt) |0.890|2.92|1.7725|
> |AsyncReasoning+ThinkSwitcher|0.932|68.2893|3.5484|
>
> Both DEER and ThinkSwitcher reduce thinking delays by either thinking less or answering some questions right away, but thinking in parallel offers even better trade-offs by overlapping thinking and writing. Crucially, these strategies can be combined with AsyncReasoning to achieve even better accuracy to speed trade-offs. As we discuss in L148-156, the only prior work on training-free streaming reasoning [6] overlaps *reading* and thinking, which would have no effect in our evaluation.
>
> We will include these new results in the final version of the paper and discuss other techniques [7–10]. Additionally, since AsyncReasoning is orthogonal to most efficient reasoning techniques, we will study these techniques in combination with ours.
>
> > 2.the effectiveness of AsyncReasoning may depend heavily on the post-training data format and the extent to which the base model has already been exposed to reasoning-style or multi-mode interaction patterns.
>
> We agree that AsyncReasoning depends on the model to be capable of reasoning in some format. Note that the two main model families (Qwen3 and GPT-OSS) in our experiments **use different formats**: Qwen3 uses `<think>blocks</think>` while GPT-OSS uses Harmony [11]. This suggests that AsyncReasoning can generalize between different reasoning formats.
>
> However, our method requires **LLMs with strong reasoning capabilities** (L23 onward). To examine this limitation, we evaluated AsyncReasoning on **Llama-3.1-8B-Instruct** using **MATH-500**. The non-thinking baseline achieved an accuracy of **0.484**, whereas **AsyncReasoning** obtained **0.412**. Closer inspection revealed that the non-thinking model was often distracted by the mode-switching prompt and became trapped in a repetition loop.
>
> > 3.there is still a nontrivial drop in task performance in some settings. This raises an important practical question: how should users trade off reasoning quality against responsiveness?
>
> While we agree with the premise, we would like to reaffirm that the performance drop is small for sufficiently ‘smart’ models (30B+) in many practical settings (Fig. 5 & 6), making it favorable for those models. In smaller models (0.6B, 1.7B), the drops are indeed more noticeable. We present some analysis of speed-accuracy tradeoffs in Fig. 4, Appendix E and in response to Reviewer TaQz (**Table TaQZ.2**) and will discuss this further in the final version of the paper. In future, it would be interesting to see if small models can develop better switching / pausing policies through training (as opposed to training-free AsyncReasoning) or with ad-hoc classifiers.
>
> We thank you for the suggestions and hope that our new experiments alleviated your concerns. If they did, please consider revisiting your score accordingly. We will be glad to address further suggestions and answer questions in the author-reviewer discussion phase.
>
> - [1] https://arxiv.org/abs/2504.15895
> - [2] https://arxiv.org/abs/2509.14004
> - [3] https://arxiv.org/abs/2505.14183
> - [4] https://arxiv.org/abs/2503.05179
> - [5] https://arxiv.org/abs/2504.13367
> - [6] https://arxiv.org/abs/2510.17238
> - [7] https://arxiv.org/abs/2401.05618
> - [8] https://arxiv.org/abs/2412.18547
> - [9] https://arxiv.org/abs/2502.18600
> - [10] https://arxiv.org/abs/2503.16419
> - [11] https://github.com/openai/harmony

---

> > ### Author Rebuttal · Reviewer_gTPM · 2026-04-03
> >
> > Thank you for the new experimental results. I will raise to weak accept

---

> > > ### Author Response · Authors · 2026-04-07
> > >
> > > Thank you for the update and for your suggestions! We are glad that the discussion addressed your concerns.

---

### Official Review · Reviewer_9yTv · 2026-03-16

**Soundness:** 2
**Presentation:** 3
**Significance:** 3
**Originality:** 3
**Overall Recommendation:** 4
**Confidence:** 3

**Summary:**

The paper introduces AsyncReasoning, a training-free method that enables reasoning LLMs to think and respond concurrently by maintaining separate Thinker and Writer token streams over a shared KV cache. The key mechanism exploits the geometric properties of RoPE to make the same cached tokens appear in different sequential positions depending on the stream, without physically rearranging memory. A periodic mode-switching prompt lets the model decide whether to pause writing and wait for more reasoning. The method reduces time-to-first-token to ≤5s and user-perceived silence time by up to 12× compared to standard sequential reasoning, with accuracy broadly preserved on selected benchmarks.

**Compliance With Llm Reviewing Policy:**

Affirmed.

**Key Questions For Authors:**

- The paper falls back to the simplest mode-switching prompt after observing erratic behavior on GPQA-Diamond with more advanced strategies. But the simplest prompt is a fixed-interval heuristic — it does not adapt to task difficulty or model capability. Is there a principled criterion (e.g., uncertainty in the Thinker, entropy of the next Writer token) that could make mode switching more reliable without requiring training?

- The safety evaluation with Safety Prompt 2 blocks the Writer for the first 1024 Thinker tokens, making the initial phase effectively synchronous. What is the ASR and accuracy when the Writer is allowed to run concurrently from the start (i.e., without the 1024-token blocking period)? Without this condition, it is unclear whether the async architecture itself contributes to the safety gain or whether the result is driven entirely by the safety prompt content.

- SpokenMQA is the only benchmark that directly evaluates the voice assistant use case motivating the paper, yet it is reported only for Qwen3-0.6B and 4B. Why were the larger models (Qwen3-32B, 235B-A22B, GPT-OSS) not evaluated on SpokenMQA? Given that production voice assistants are typically backed by large models, is the omission due to a computational constraint, a dataset limitation, or an accuracy concern?

- Appendix I identifies race conditions as a failure mode where the Writer outputs harmful content before the Thinker can intervene. While the Thinker attends to Writer tokens and is therefore aware of what is being generated, the mode-switching mechanism only gates whether writing should start or pause — there is no mechanism for the Thinker to interrupt or halt the Writer mid-stream upon detecting harmful content already being produced. Is there a way to extend AsyncReasoning with such an interruption mechanism?

**Limitations:**

yes

**Strengths And Weaknesses:**

## Strengths

**The positional embedding manipulation is technically elegant and genuinely training-free.**
The core trick — reusing a single KV cache with per-query RoPE rotations to simulate two different token orderings — is clean and avoids the common pitfall of requiring fine-tuning or architectural changes. The method slots into existing inference frameworks with only a custom attention kernel.

**The real-time delay metric is well-motivated and appropriate for the target application.**
Rather than relying solely on TTFT or throughput, the paper defines "silence time" — cumulative time the TTS engine has nothing to speak — which directly captures user-perceived latency in voice assistant scenarios. This is a more ecologically valid metric than standard benchmarks typically use.


## Weaknesses

**Mode switching is fragile and model-dependent.**
The paper acknowledges that advanced prompting strategies (thinker async, switch prompt) "behave erratically on some benchmarks, notably GPQA-Diamond." The authors' response is to fall back to the simplest prompt, which sidesteps rather than solves the problem. Since mode switching is the central mechanism controlling the accuracy-latency tradeoff, its sensitivity to prompt choice raises concerns about reliability across tasks and models not covered in the evaluation.

**The safety benefit is likely attributable to prompting, not the async architecture.**
The paper claims AsyncReasoning enables safety screening via the background Thinker, reducing ASR from 12.5% to 0.5%. However, Safety Prompt 2 blocks the Writer for the first 1024 tokens entirely — making this phase effectively synchronous. The strong safety result is more plausibly explained by careful safety prompting than by the async mechanism itself. A necessary ablation is missing: a baseline that injects safety prompts mid-thinking in a standard sequential or interleaved setup — analogous to how the interactive baseline works for latency — would isolate whether the async separation of Thinker and Writer is actually responsible for the safety gain, or whether periodic safety prompting alone suffices. The accuracy drop (0.94 → 0.77) under Safety Prompt 2 further suggests aggressive prompting is doing the heavy lifting.

**Benchmark coverage is inconsistent and underreports the most relevant use case.**
Model-benchmark coverage is uneven across the appendix tables. MATH-500 omits Qwen3-235B-A22B and GPT-OSS entirely; GPQA-Diamond omits GPT-OSS; AIME-2025 is limited to just two models. Most critically, SpokenMQA — the only benchmark directly targeting the voice assistant use case that motivates the entire paper — is evaluated on only two small models (Qwen3-0.6B and 4B). The largest and most capable models, which are most likely to be deployed in real voice assistant products, are absent from the benchmark most relevant to the paper's core motivation.

---

> ### Author Rebuttal · Authors · 2026-03-30
>
> Thank you for your detailed feedback and suggestions. We are glad that you appreciate the technical elegance and the real-time delay evaluation. Below, we provide additional evaluations and clarifications to address your concerns.
>
> > Mode switching is fragile and model-dependent. (...) Is there a principled criterion (e.g., uncertainty in the Thinker, entropy of the next Writer token) that could make mode switching more reliable without requiring training?
>
> Our main assumption is that if we ask the mode-switching question often enough (we use T=20, i.e. about once per line of text on A4), it will be responsive enough without incurring significant overhead - and it was enough in our experiments. That said, we agree that there can be other use cases where more frequent mode switching is valuable. Using uncertainty / entropy in model predictions could be useful if the entropy is meaningful, but there is a caveat: many instruct- and reasoning-tuned LLM families are overconfident in terms of next token probability[1]. An alternative direction is to train an ad-hoc classifier to determine when to ask the mode-switching question.
>
>
> > The safety benefit is likely attributable to prompting, not the async architecture. (...) What is the ASR and accuracy when the Writer is allowed to run concurrently from the start (i.e., without the 1024-token blocking period)?
>
> We agree that the 1024-token delay was a significant limitation for safety experiments. Since the submission, we systematically experimented with more general safety reasoning without hard-coded delays. What ended up working is using specialized prompts (both initial and during mode switching) to guide the model when it should pause for safety reasons. Below, we report both the original prompt without 1024 token delay, synchronous reasoning about safety, and the new prompts *without the 1024 delay*.
>
> ### Table 9yTv.1: New safety evaluations, Qwen3-32B
> |Setup|ASR on harmbench|Accuracy on MATH-500|
> |-|-|-|
> |Synchronous w/o safety prompt|12.5%|0.94|
> |Synchronous w/ safety prompt|0%|0.93|
> |Ours (old w/ delay)|0.5%|0.93|
> |Ours (old w/o delay)|2%|0.89|
> |Ours (new w/o delay)|0.5%|0.93|
>
> Both old and new prompts have comparable ASRs. New prompt is simpler in its design and works better without the 1024 token delay. Synchronous reasoning w/ safety prompts does protect against the attacks significantly increasing delays. AsyncReasoning allows the model to work interactively, which is important in practice. It also preserves accuracy on MATH-500.
>
> > Model-benchmark coverage is uneven across the appendix tables. MATH-500 omits Qwen3-235B-A22B and GPT-OSS entirely; GPQA-Diamond omits GPT-OSS; AIME-2025 is limited to just two models.
>
> We understand the concern. Though AIME is reported for **four models (see Table 12)**, we agree that the current benchmark coverage feels uneven. The reason we chose the original layout is to save compute, e.g. by not running the large 235B model on relatively simple benchmarks (e.g. MATH-500) where its reasoning might not be fully utilized. However, we did not explain that in the submission, and you are correct that additional evaluations would strengthen our work. Below we report both Qwen3-235B on MATH-500 and smaller models on AIME-2025.
>
> ### Table 9yTv.2: Qwen3-235B-A22B-Thinking-2507 on MATH-500
> | |Accuracy|Avg. Delay (s.)|TTFT (s.)|
> |-|-|-|-|
> |Baseline (Non-Thinking)|0.928|6.519|6.519|
> |Baseline (Thinking)|0.966|1675.66|1675.66|
> |AsyncReasoning|0.962|159.198|9.979|
>
> ### Table 9yTv.3: AIME-2025 with smaller Qwen3 models
> |Qwen3|Baseline(Non-Thinking)|Baseline(Thinking)|AsyncReasoning|
> |-|-|-|-|
> |0.6B|0.023|0.15|0.147|
> |1.7B|0.096|0.287|0.267|
> |4B|0.177|0.467|0.407|
> |8B|0.207|0.453|0.433|
> |14B|0.197|0.49|0.493|
> |32B|0.2|0.533|0.493|
>
> > Why were the larger models (Qwen3-32B, 235B-A22B, GPT-OSS) not evaluated on SpokenMQA?
>
> ### Table 9yTv.4: Accuracy on SpokenMQA multi_step_reasoning evaluations with larger models
> |Qwen3|Baseline(Non-Thinking)|Baseline(Thinking)|AsyncReasoning|
> |-|-|-|-|
> |4B|80.7|81.5|81.0|
> |8B|81.2|81.5|82.4|
> |14B|82.7|84.2|84.0|
> |30B-A3B|83.8|83.1|82.7|
> |32B|83.2|84.2|84.4|
>
> The above results show an overall similar trend to our main evaluations, with AsyncReasoning closely tracking synchronous reasoning accuracy. However, on **SpokenMQA, large models do not always benefit from long reasoning**. This is because the benchmark was developed for speech-minded realtime LLMs that mostly did not use reasoning. Upon inspection, these problems turn out to be much easier than even MATH-500 questions (see [2, 3]).
>
> If our new results help address your concerns, please consider updating your score. We will also be glad to continue the discussion and address any follow-up questions and suggestions in the discussion phase.
>
> - [1] https://arxiv.org/abs/2409.19817
> - [2] https://huggingface.co/datasets/amao0o0/spoken-mqa/viewer/default/multi_step_reasoning
> - [3] https://huggingface.co/datasets/HuggingFaceH4/MATH-500/viewer/default/test

---

> > ### Author Rebuttal · Reviewer_9yTv · 2026-04-03
> >
> > Thank you for running the new experiments. I'm leaning towards accept.

---

> > > ### Author Response · Authors · 2026-04-07
> > >
> > > Thank you for the thorough review! The safety analysis and model-benchmark evaluations are a great addition to our work.
> > >
> > > P.S. If by “leaning towards accept” you mean a firm accept rather than a weak accept, we would be very grateful if you would consider updating the score accordingly.

---

### Decision · Program_Chairs · 2026-04-30

**Decision:**

Reject

**Comment:**

There is some discrepancy among the current scores. I have briefly reviewed the paper, along with the reviews and the authors’ rebuttal. In my view, most of the concerns raised by reviewer frKM (as well as those from other reviewers) have been reasonably addressed.

However, I remain somewhat unconvinced about the level of novelty. The use of rotational positional embeddings to manipulate sequence structure is a relatively well-established technique, and the primary contribution of this work appears to be its application to reducing interaction latency. While this is a practically relevant direction, it does not seem to constitute a sufficiently strong conceptual or technical advance.

Overall, I consider this a borderline submission. Given the limited novelty, I am slightly inclined toward rejection.